# A D-2-hydroxyglutarate dehydrogenase mutant reveals a critical role for ketone body metabolism in *Caenorhabditis elegans* development

Olga Ponomarova[1], Hefei Zhang[1], Xuhang Li[1], Shivani Nanda[1], Thomas B. Leland[1], Bennett W. Fox[2], Alyxandra N. Starbard[1], Gabrielle E. Giese[1], Frank C. Schroeder[2], L. Safak Yilmaz[1], Albertha J. M. Walhout[1] *

1 Department of Systems Biology, University of Massachusetts Chan Medical School, Worcester, Massachusetts, United States of America, 2 Boyce Thompson Institute and Department of Chemistry and Chemical Biology, Cornell University, Ithaca, New York, United States of America

* marian.walhout@umassmed.edu

**Data Availability Statement:** RNA-seq data available at GSE201645.

## Abstract

In humans, mutations in D-2-hydroxyglutarate (D-2HG) dehydrogenase (D2HGDH) result in D-2HG accumulation, delayed development, seizures, and ataxia. While the mechanisms of 2HG-associated diseases have been studied extensively, the endogenous metabolism of D-2HG remains unclear in any organism. Here, we find that, in *Caenorhabditis elegans*, D-2HG is produced in the propionate shunt, which is transcriptionally activated when flux through the canonical, vitamin B12-dependent propionate breakdown pathway is perturbed. Loss of the D2HGDH ortholog, *dhgd-1*, results in embryonic lethality, mitochondrial defects, and the up-regulation of ketone body metabolism genes. Viability can be rescued by RNAi of *hphd-1*, which encodes the enzyme that produces D-2HG or by supplementing either vitamin B12 or the ketone bodies 3-hydroxybutyrate (3HB) and acetoacetate (AA). Altogether, our findings support a model in which *C. elegans* relies on ketone bodies for energy when vitamin B12 levels are low and in which a loss of *dhgd-1* causes lethality by limiting ketone body production.

## Highlights

- D-2-hydroxyglutarate is produced by HPHD-1 in the propionate shunt pathway.
- DHGD-1 recycles 2-hydroxyglutarate to sustain flux through the propionate shunt.
- *dhgd-1* loss perturbs ketone body metabolism and causes embryonic lethality.
- 3-Hydroxybutyrate, acetoacetate, vitamin B12, or *hphd-1* RNAi rescue *dhgd-1* mutant lethality.

**Funding:** This work was supported by grants GM122502 and DK068429 from the National Institutes of Health to A.J.M.W. and DK115690 to A.J.M.W. and F.S., and by a grant from the Li Weibo Institute for Rare Disease at University of Massachusetts Chan Medical School to A.J.M.W. The funders had no role in study design, data collection and analysis, decision to publish, or preparation of the manuscript.

**Competing interests:** The authors have declared that no competing interests exist.

**Abbreviations:** AA, acetoacetate; BCAA, branch chain amino acid; CGC, Caenorhabditis Genetics Center; D-2HG, D-2-hydroxyglutarate; D2HGDH, 2-hydroxyglutarate dehydrogenase; FBA, flux balance analysis; FDR, false discovery rate; GC-MS, gas chromatography–mass spectrometry; GSEA, gene set enrichment analysis; HMB, hydroxymethylbutyrate; MS, methionine synthase; MSA, malonic semialdehyde; NGM, nematode growth medium; SCFA, short chain fatty acid.

## Introduction

The metabolite 2HG occurs as 2 enantiomers, L-2HG and D-2-hydroxyglutarate (D-2HG), each of which is oxidized by a specific dehydrogenase. Mutations in these dehydrogenases result in the inborn errors of human metabolism, L- and D-2-hydroxyglutaric aciduria, respectively [1,2]. These diseases cause the accumulation of 2HG in bodily fluids, delayed development, neurological and muscle dysfunction, and early death [3]. Both enantiomers are oncometabolites, but they are produced differently and have distinct effects on metabolism and physiology. L-2HG accumulates during hypoxia and is produced by malate and lactate dehydrogenases [4,5]. Both L-2HG and D-2HG accumulate in humans with mutated mitochondrial citrate transporter [6], and an underlying mechanism of this disorder was proposed by studies in model organism *Drosophila* [7]. D-2HG drives oncogenic transformation in cells with neomorphic mutations in the isocitrate dehydrogenases IDH1 and IDH2 [8,9]. D-2HG inhibits multiple enzymes, including alpha-ketoglutarate (αKG)-dependent dioxygenases [10,11], BCAT transaminases [12], αKG dehydrogenase [13], and ATP synthase [14]. D-2HG can be produced by several enzymes, including the hydroxyacid-oxoacid transhydrogenase ADHFE1 [15], the phosphoglycerate dehydrogenase PHGDH [16], and wild-type IDH1 and IDH2 [5]. However, it remains unclear if D-2HG production bears any functional significance or is due to promiscuous enzyme activity.

In eukaryotes, the degradation of the branch chain amino acids (BCAAs) valine and isoleucine yields propionyl-CoA, which can be converted to the short chain fatty acid (SCFA) propionate. In addition, propionate is produced by the gut microbiota during the breakdown of dietary fiber [17]. Together with the other major SCFAs, acetate and butyrate, propionate forms an important source of energy for muscle, colon, and liver [18,19]. Thus, propionate serves an important metabolic and physiological function. However, propionate is toxic when it accumulates to high levels in the blood, which occurs in patients with propionic acidemia that carry mutations in either of the 2 propionyl-CoA carboxylase subunits [20]. These enzymes function together in the canonical, vitamin B12-dependent propionyl-CoA breakdown pathway that leads to the production of succinyl-CoA, which can anaplerotically enter the TCA cycle to produce energy [21].

Ketone bodies provide another important energy source under conditions where glucose is limiting, such as in diabetic patients, or on low carbohydrate, or "keto" diets. Ketone bodies are produced in the liver and are essential for energy metabolism in peripheral tissues such as muscle and the brain. There are 3 ketone bodies, 2 of which, acetoacetate (AA) and 3-hydroxybutyrate (3HB), can serve as energy sources, with 3HB being the most prevalent. Ketone bodies are produced from the breakdown of fatty acids and amino acids. Two amino acids, lysine and leucine, are exclusively ketogenic: degradation of leucine yields acetyl-CoA and acetoacetate, while breakdown of lysine yields 3HB. Furthermore, acetyl-CoA produced by leucine (and fatty acids) is an important precursor for ketone body production.

The nematode *Caenorhabditis elegans* has been a powerful model organism for decades, and research on this "simple" animal has yielded great insights into development, aging, and other processes. Recently, *C. elegans* has also become a major model to study metabolism. Of its 20,000 or so protein-coding genes, more than 2,000 are predicted to encode metabolic enzymes. Of these, 1,314 have been annotated to specific metabolic reactions and incorporated into a genome-scale metabolic network model that can be used with flux balance analysis (FBA) to computationally model the animal's metabolism [22,23]. Like in humans, vitamin B12 plays an important metabolic role in *C. elegans* propionate breakdown [24–27]. Vitamin B12 is exclusively synthesized by bacteria and, perhaps, some archaea, and is therefore mostly acquired by diet. On bacterial diets low in vitamin B12, such as the standard laboratory diet of

*Escherichia coli* OP50, *C. elegans* transcriptionally activates 5 genes that comprise an alternative propionate breakdown pathway or propionate shunt [27–29]. This shunt is activated only when high levels of propionate persist and detoxifies this SCFA to acetyl-CoA in the mitochondria [28].

Here, we use a loss-of-function mutation in *C. elegans* D-2-hydroxyglutarate dehydrogenase (D2HGDH), which converts D-2HG to αKG, to study endogenous D-2HG metabolism. We identified the *C. elegans* gene F54D5.12 as a one-to-one ortholog of human D2HGDH and named this gene *dhgd-1* for D-2-hydroxyglutarate dehydrogenase. We find that D-2HG is produced by the propionate shunt, in the step in which HPHD-1 oxidizes 3-hydroxypropionate (3HP) to malonic semialdehyde (MSA). *dhgd-1* deletion mutants are embryonic lethal and have mitochondrial defects. Surprisingly, however, while mitochondrial defects can be explained by 3HP accumulation [30], lethality is neither simply caused by accumulation of 3HP nor D-2HG. Therefore, loss of viability and mitochondrial defects are caused by distinct mechanisms and can be uncoupled. We find that *dhgd-1* mutant animals up-regulate ketone body metabolism genes, suggesting that ketone bodies are limiting in these animals. Indeed, our metabolomic analysis shows that the breakdown of the ketogenic amino acids leucine and lysine is impaired in these mutants. Moreover, the ketone bodies 3HB and AA can partially rescue *dhgd-1* mutant lethality. Altogether, our findings support a model in which ketone bodies are important for *C. elegans* viability.

## Results

### DHGD-1 is a D-2-hydroxyglutarate dehydrogenase

We first asked whether DHGD-1 is indeed a D-2HG dehydrogenase (**Fig 1A**). DHGD-1 sequence is 46% identical to its one-to-one human ortholog D2HGDH and contains the same conserved functional domains (**S1A** and **S1B Fig** and **S1 Table**). We obtained *dhgd-1(tm6671)* mutant animals that carry a large deletion in the 5′ end of the gene, which is predicted to result in loss of gene function (**Fig 1B**). These animals are hereafter referred to as Δ*dhgd-1* mutants. By gas chromatography–mass spectrometry (GC-MS), we found that Δ*dhgd-1* mutants accumulate about 5-fold more 2HG than wild-type animals (**Fig 1C**). Human D2HGDH specifically oxidizes the D-form of 2HG to αKG [31]. To test the stereo-specificity of *C. elegans* DHGD-1, we used chiral derivatization that allows the chromatographic separation of the D- and L-2HG enantiomers. While in wild-type animals, the ratio between L- and D-2HG is about one to one, Δ*dhgd-1* mutant animals predominantly accumulate the D-2HG enantiomer, confirming that DHGD-1 is indeed a functional ortholog of human D2HGDH (**Fig 1D**).

We found that Δ*dhgd-1* mutant animals have severe embryonic lethality as approximately two thirds of the embryos failed to hatch (**Fig 1E and 1F**). In addition, Δ*dhgd-1* mutant animals exhibit mitochondrial defects as their mitochondrial network is more fragmented and individual mitochondria are more rounded, which agrees with recently reported data [30] (**S2A Fig**). These observations allow us to investigate the connection between organismal physiology and endogenous D-2HG metabolism.

### DHGD-1 functions in the propionate shunt

The Δ*dhgd-1* mutant metabolome we measured by GC-MS revealed several additional changes in metabolite levels in adults, including lower abundance of glutamate, αKG, succinate, and the lysine breakdown products 2-aminoadipate (2AA) and glutarate, as well as higher abundance of lysine, β-alanine, 3-aminoisobutyrate, and the leucine breakdown product 3,3-hydroxymethylbutyrate (HMB) (**Fig 2A**). These results suggest that the breakdown of the ketogenic amino acids leucine and lysine is impaired in these mutants. Importantly, 3HP, a metabolite

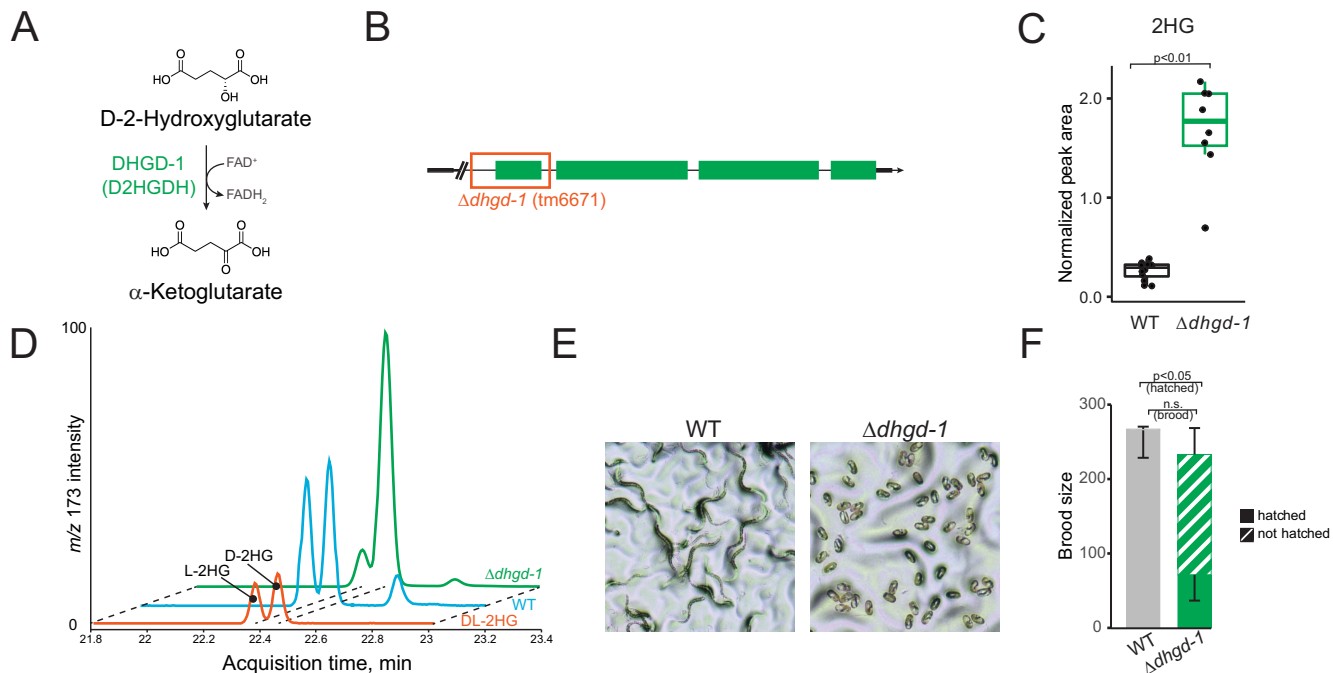

**Fig 1. *Dhgd-1* encodes a D-2HG dehydrogenase.** (A) Metabolic reaction catalyzed by D-2HG dehydrogenase. (B) Schematic of deletion in *dhgd-1*(*tm6671*) mutants, referred to as *Δdhgd-1* mutants. (C) GC-MS measurement of 2HG accumulation in *Δdhgd-1* mutants and WT animals. Each dot represents an independent biological replicate. (D) Discrimination between D- and L-2HG enantiomers by chiral GC-MS derivatization in WT and *Δdhgd-1* mutant animals. Raw data are not normalized and therefore do not reflect 2HG concentration in different samples. (E, F) Embryonic lethality and brood size in *Δdhgd-1* mutants. Brightfield images (E) and hatching (F). Scale bar 50 μm. Bars in (F) represent mean and standard deviation of *n* = 3 biological replicates. n.s.–not significant. The data underlying **Fig 1C and 1F** can be found in **S1 Data**. D-2HG, D-2-hydroxyglutarate; GC-MS, gas chromatography–mass spectrometry; WT, wild type.

unique to the propionate shunt [27], exhibited the greatest increase in abundance in *Δdhgd-1* mutant animals, which suggests that loss of *dhgd-1* interferes with the function of the propionate shunt. Increased abundance of 2HG and 3HP was also evident in *Δdhgd-1* embryos and L4 larval stage animals (**S3A and S3B Fig**).

By using a compendium of publicly available *C. elegans* expression profiles from multiple growth conditions and genetic backgrounds, we calculated pairwise correlation in expression of *dhgd-1* with other metabolic genes [32]. We found that 4 of the 5 propionate shunt genes are strongly coexpressed with *dhgd-1* (**Fig 2B and 2C** and **S2 Table**) One of these, *hphd-1*, is an ortholog of human ADHFE1, which produces D-2HG in a reaction coupled to oxidation of the neurotransmitter and psychoactive drug γ-hydroxybutyrate (GHB) [15]. Therefore, we hypothesized that HPHD-1 reduces αKG to D-2HG when it oxidizes 3HP to MSA [27] and that DHGD-1 recycles D-2HG back to αKG (**Fig 2D**). Since HPHD-1 harbors a highly conserved Rossman fold that can bind nucleic acid cofactors such as NAD+, we predicted that HPHD-1 uses NAD+/NADH to shuttle a hydride from 3HP to D-2HG (**Fig 2E**) [33]. If true, this leads to the prediction that D-2HG is derived from propionate degradation. To directly test this, we performed a stable isotope tracing experiment in which animals were supplemented with either propionate or deuterated $^2H_5$-propionate. Supplementing $^2H_5$-propionate produced both deuterated 3HP and D-2HG, while the bacterial diet alone did not contain any detectable 3HP or 2HG, showing that these conversions happen in the animal (**Figs 2F–2H** and **S4**). This result demonstrates that production of D-2HG is coupled to oxidation of 3HP in the propionate shunt. We asked whether GHB could be an alternative source of D-2HG but

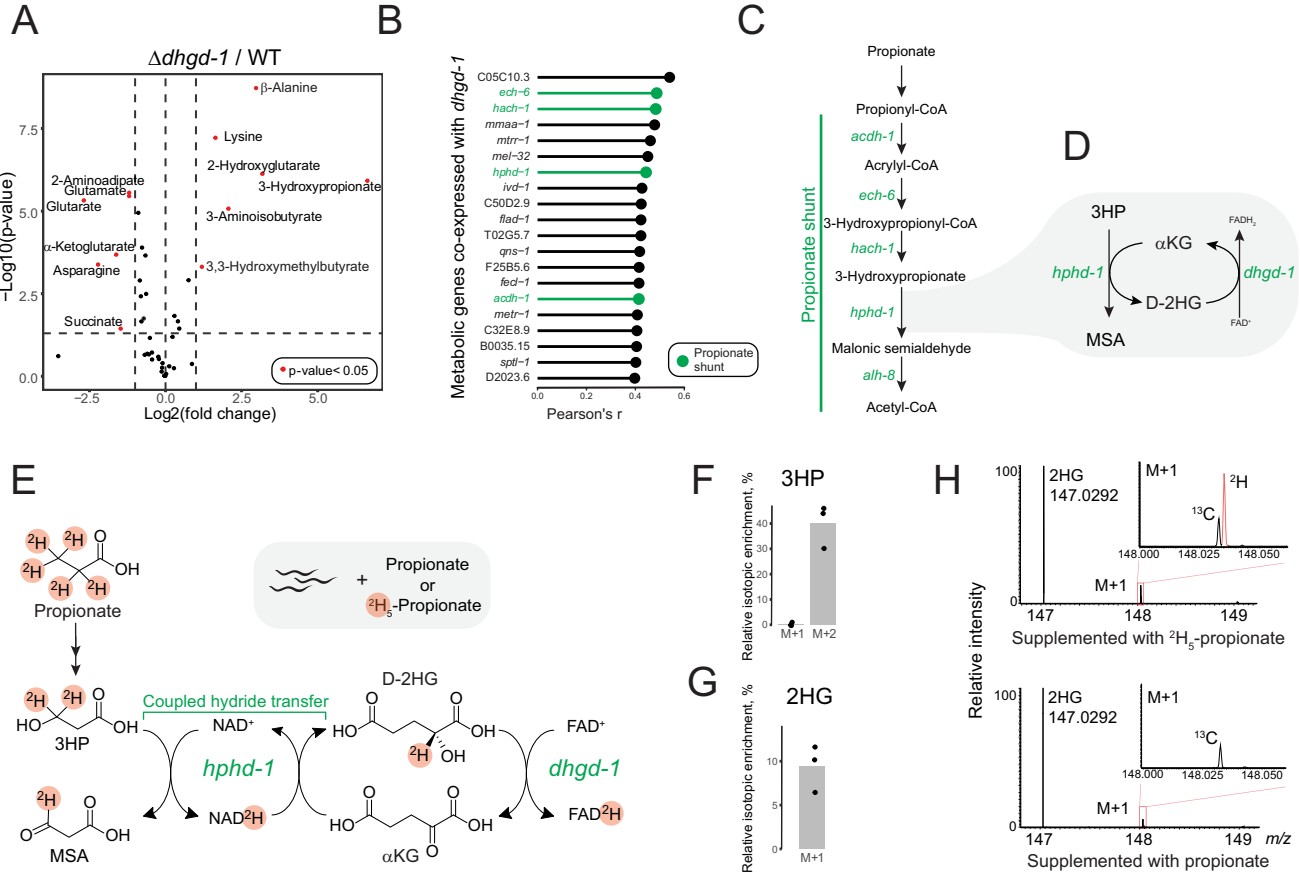

**Fig 2. *Dhgd-1* functions in the propionate shunt pathway.** (A) GC-MS profiling of metabolic changes in *Δdhgd-1* mutants compared to WT *C. elegans*. *P*-values are Benjamini–Hochberg adjusted. (B) Metabolic genes most highly coexpressed with *dhgd-1*. Propionate shunt pathway genes are enriched with an FDR of 0.042 (S2 Table). (C, D) Proposed mechanism of D-2HG production (by HPHD-1) and recycling (by DHGD-1) during propionate degradation via the propionate shunt pathway. (E) Isotope tracing experiment with deuterium ($^2$H)-labeled propionate and proposed reaction mechanism. HPHD-1 uses NAD$^+$/NADH as a hydride shuttle in a coupled reaction yielding 1 equivalent each of MSA and D-2HG. (F, G) Fractional enrichment of 3HP (F) and 2HG (G) isotopologues. Bars indicate mean of $n = 3$ biological replicates. (H) HPLC-MS analysis of the M+1 isotope cluster in animals fed $^2$H$_5$-propionate revealed robust incorporation of a single deuterium atom in 2HG, whereas the M+1 in animals fed propionate was exclusively from natural abundance of $^{13}$C. The data underlying Fig 2F and 2G can be found in S1 Data. D-2HG, D-2-hydroxyglutarate; FDR, false discovery rate; GC-MS, gas chromatography–mass spectrometry; HPLC-MS, high performance liquid chromatography mass spectrometry; MSA, malonic semialdehyde; WT, wild type.

could not detect this metabolite in the animal (S1C Fig). We therefore conclude that DHGD-1 functions in the propionate shunt, and that its dysfunction leads to impaired flux through this pathway, resulting in accumulation of shunt intermediates 3HP and 2HG, as well as mitochondrial dysfunction and embryonic lethality.

## *hphd-1* RNAi and vitamin B12 supplementation rescue embryonic lethality of *Δdhgd-1* mutants

High levels of 3HP cause mitochondrial defects in *Δdhgd-1* mutants [30]. We therefore wondered whether these defects cause embryonic lethality. To test this, we performed RNAi knock-down of *hphd-1*, which we predicted to reduce 2HG but not 3HP levels (Fig 3A). Indeed, RNAi of *hphd-1* reduced 2HG to levels that are similar to wild-type animals but did not affect 3HP levels (Fig 3B). Remarkably, RNAi of *hphd-1* almost fully rescued lethality of *Δdhgd-1* mutants (Fig 3C). This result shows that embryonic lethality can be uncoupled from

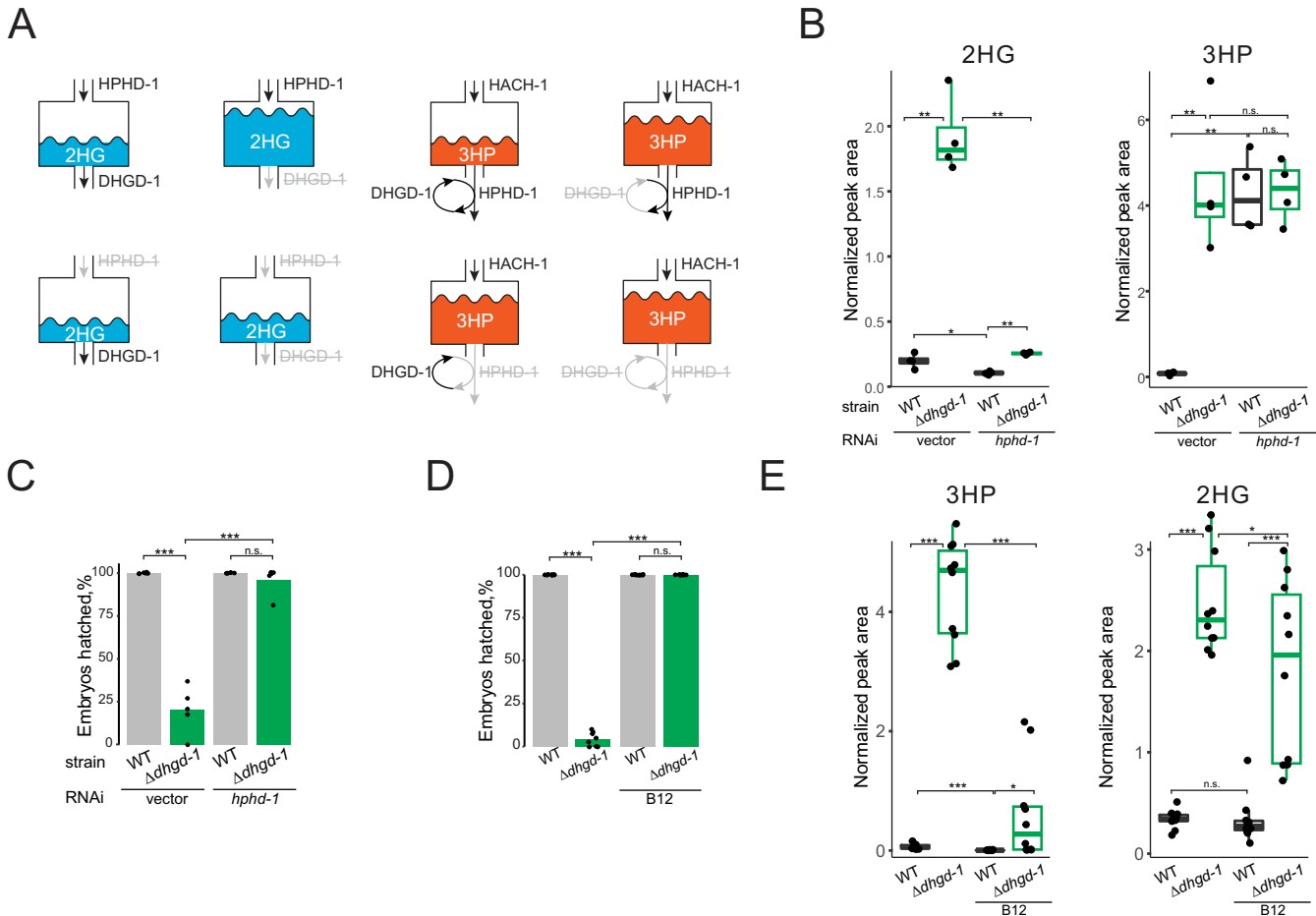

**Fig 3. Rescue of lethality in *Δdhgd-1* mutants by vitamin B12 supplementation and *hpdh-1* RNAi.** (A) Schematic of predicted HPHD-1 and DHGD-1 contributions to 2HG (blue) and 3HP (orange) accumulation. HPHD-1 is a main source of 2HG, and therefore, its knockdown prevents 2HG accumulation in case of DHGD-1 dysfunction. 3HP is expected to accumulate if either DHGD-1 or HPHD-1 is perturbed because these reactions are coupled to facilitate 3HP oxidation. (B) 2HG and 3HP abundance in *Δdhgd-1* mutants upon RNAi of *hphd-1*. Boxplot midline represents median of 4 independent biological replicates (dots). (C) *hphd-1* RNAi rescues lethality in *Δdhgd-1* mutants. The RNAi-compatible *E. coli* OP50 strain [34] was used because conventional RNAi-compatible *E. coli* HT115 rescued embryonic lethality of *Δdhgd-1* animals. Each dot represents an independent biological replicate and bars indicate means. (D) Vitamin B12 rescues lethality in *Δdhgd-1* mutants. Each dot represents an independent biological replicate and bars indicate means. (E) 3HP and 2HG abundance in *Δdhgd-1* mutants supplemented with vitamin B12. Boxplot midline represents median of independent biological replicates (dots). All panels: The means of 3 or more groups were compared with ANOVA, followed by unpaired *t* test (\**p* < 0.05, \*\**p* < 0.01, \*\*\**p* < 0.001). The data underlying **Fig 3B–3E** can be found in **S1 Data**. WT, wild type.

mitochondrial defects, which are not rescued by *hphd-1* perturbation [30]. Both *dhgd-1* mutation and *hphd-1* RNAi block HPHD-1 function [30]. However, *hphd-1* RNAi rescues embryonic lethality in *Δdhgd-1* mutants and is not lethal in wild-type animals. Therefore, we conclude that lack of flux through the propionate shunt does not explain embryonic lethality in *Δdhgd-1* mutant animals.

Flux through the propionate shunt is transcriptionally repressed by vitamin B12, which enables flux through the canonical propionate degradation pathway [26–28]. We found that supplementation of vitamin B12 rescued both mitochondrial defects and embryonic lethality in *Δdhgd-1* mutants (**Figs 3D and S2A**). Surprisingly, however, while 3HP levels went down in *Δdhgd-1* mutants upon supplementation of vitamin B12, 2HG levels decreased very little (**Fig 3E**). This observation suggests that there is still some flux through the propionate shunt pathway in *Δdhgd-1* mutant animals supplemented with vitamin B12. More importantly, this result

indicates that high levels of 2HG are not sufficient to elicit embryonic lethality in these mutants. Importantly, levels of 3HP and 2HG were strongly correlated even in vitamin B12-supplemented *Δdhgd-1* animals, indicating that their metabolism is still coupled (**S5 Fig**). This coordination is even more evident on a diet of RNAi competent *E. coli* OP50 (xu363) [34] where vitamin B12 reduces the levels of both 3HP and 2HG by more than 2 folds, in adults and embryos (**S3F and S3I Fig**). Therefore, we hypothesized that vitamin B12 supplementation rescued embryonic lethality in *Δdhgd-1* mutants not by lowering 2HG accumulation but by compensating for its detrimental effects, for example, its inhibition of the activity of metabolic enzymes involved in the breakdown of leucine and/or lysine.

## Loss of *dhgd-1* may cause lethality by impairing ketone body production

To gain more insight into the mechanism by which loss of *dhgd-1* causes embryonic lethality and how this could be rescued by vitamin B12 supplementation, we used expression profiling by RNA sequencing (RNA-seq). Overall, 315 and 183 genes are induced and repressed by loss of *dhgd-1*, respectively (**S3 Table**). WormFlux [23] pathway enrichment analysis using pathways defined by WormPaths [35] revealed 2 main insights. First, we found that the propionate shunt is not fully repressed by vitamin B12 in *Δdhgd-1* mutants (**Fig 4A and 4B** and **S4 Table**). This observation suggests that the propionate shunt remains active in the presence of vitamin B12 and can explain why 2HG levels remain high in these animals (**Fig 3E**). Second, ketone body metabolism genes are up-regulated in *Δdhgd-1* mutants with or without vitamin B12 compared to wild type and are down-regulated in both *Δdhgd-1* mutants and wild type by vitamin B12 in relation to their respective genotypes (**Fig 4A and 4B**). Based on this result, we hypothesized that perturbed ketone body metabolism may explain embryonic lethality of *Δdhgd-1* mutant animals. This hypothesis is further supported by our metabolomic analysis, because we found differential accumulation of metabolites in the breakdown of the ketogenic amino acids, leucine and lysine, in *Δdhgd-1* mutants: lysine levels are increased while its breakdown products 2-aminoadipate and glutarate are decreased, and the leucine breakdown product HMB is increased (**Figs 2A, S3A, S3B, S3E and S3H**). This result suggests that the breakdown of ketogenic amino acids, lysine and leucine, in *Δdhgd-1* mutants is impaired, at initial and intermediate steps, respectively, resulting in lower levels of ketone bodies.

To test this hypothesis, we supplemented *Δdhgd-1* mutants with the ketone bodies 3HB or AA and found that either partially rescued embryonic lethality, while acetate had no effect (**Figs 4C** and **S7A**). 3HB rescue is specific to *Δdhgd-1* animals since it does not affect embryonic lethality in other mutants tested (**S7B Fig**). 3HB supplementation does not change 2HG levels in *Δdhgd-1* mutant animals (**S7C and S7D Fig**) and does not rescue their mitochondrial defect (**S7E Fig**). This is similar to rescue by vitamin B12 and supports the idea that lowering 2HG levels is not required to mitigate embryonic lethality. Together, these results support the conclusion that impaired ketone body metabolism (**Fig 4D**) causes embryonic lethality in *Δdhgd-1* mutant animals.

Why are ketone bodies required for viability in *Δdhgd-1* mutant animals? In mammals, ketone bodies are important carriers of energy [36]. Therefore, we hypothesized that *dhgd-1* may be required for energy production. Since lysine and leucine breakdown are impaired in *Δdhgd-1* mutants, we modeled the impact of the combined loss of *dhgd-1* function with impaired lysine and leucine degradation on the production of both ketone bodies and energy. Specifically, we used FBA with the genome-scale *C. elegans* metabolic network model iCEL1314 and used maximization of ketone body or energy production as an objective function [22,23]. FBA predicted that energy and ketone body production are not affected by lack of vitamin B12 (loss of canonical propionate degradation pathway) or *dhgd-1* mutation (loss of

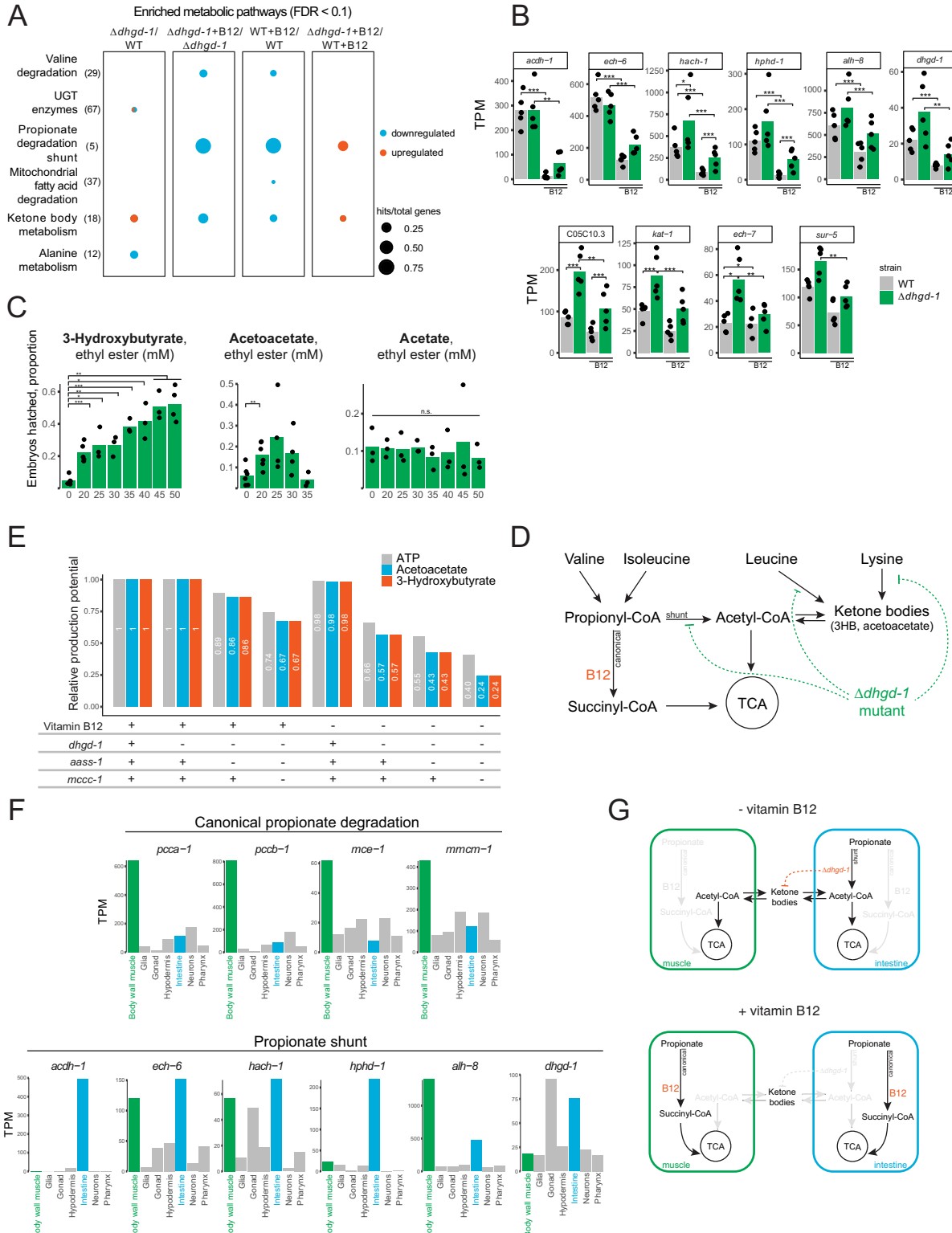

**Fig 4. Loss of *dhgd-1* activates the expression of ketone body metabolism genes.** (A) Gene expression pathway enrichment analysis for animals with and without vitamin B12 (B12). (B) RNA-seq data for propionate shunt (top) and ketone body metabolism genes (bottom). TPM–transcripts per million. Gene categories were annotated in [35]. (C) Supplementation of 3HB or AA ethyl esters partially rescues lethality of *Δdhgd-1* mutant animals. Acetate ethyl ester was used as a negative control. (D) Model for the effect of *dhgd-1* mutation on ketone body production. (E)

Contribution of propionate shunt (*dhgd-1*), lysine (*aass-1*) and leucine (*mccc-1*) degradation pathways and vitamin B12 to the generation of energy (ATP) and ketone bodies (3HB and acetoacetate). Metabolite production potentials were estimated by maximizing their output fluxes in corresponding FBA formulations. (F) Tissue expression of genes comprising the canonical vitamin B12-dependent propionate degradation pathway and propionate shunt pathway from a published single-cell RNA-seq dataset [39]. (G) Model for ketone body exchange between *C. elegans* intestine and muscle in the presence and absence of vitamin B12. All panels: the means of 3 or more groups were compared with ANOVA, followed by unpaired *t* test (\*$p < 0.05$, \*\*$p < 0.01$, \*\*\*$p < 0.001$). Sequencing data underlying **Fig 4A and 4B** have been deposited in GEO under accession code GSE201645. The data underlying **Fig 4B, 4C, 4E, and 4F** can be found in **S1 Data**. FDR, false discovery rate; WT, wild type.

the shunt pathway) alone (**Fig 3E**). However, on a diet low in vitamin B12, FBA predicted that loss of *dhgd-1* reduced both ketone body and energy production (**Fig 4E**). This result indicates that the 2 propionate breakdown pathways may be important sources of energy. Indeed, *Δdhgd-1* embryos have depleted levels of the TCA cycle metabolites malate, fumarate, and succinate (**S3A and S3E Fig**). Moreover, levels of all 3 metabolites were restored by vitamin B12 supplementation (**S3C and S3F Fig**). Importantly, the modeling predicted that the additional loss of lysine and leucine degradation exacerbated the reduction in ketone body and energy production (**Fig 4E**). This observation provides further support for our hypothesis that ketone body production is impaired in *Δdhgd-1* mutant animals (**Fig 4D**). Additionally, FBA modeling indicates that ketone bodies, derived from leucine and lysine degradation, are an important source of energy in *C. elegans* and that the propionate shunt plays a key role in ketone body production on a bacterial diet low in vitamin B12.

To test the hypothesis that lysine is an energy source in *C. elegans* and that its breakdown is impaired in *Δdhgd-1* mutant animals, we performed an isotope tracing experiment with ¹³C-labeled lysine. Label incorporation into the lysine degradation intermediate 2AA, a metabolite that is depleted in the *Δdhgd-1* mutant, confirmed that *C. elegans* catabolize lysine via this pathway (**S8 Fig**). Unfortunately, we could not test if lysine-derived carbon was incorporated into 3HB due to its low abundance and co-elution with its isomer 3-hydroxyisobutyrate. Nevertheless, we were able to detect ¹³C enrichment in citrate/isocitrate (**S8 Fig**), TCA cycle metabolites that are the closest quantifiable intermediate in ketone body oxidation [37]. Importantly, isotopic labeling of citrate was reduced in *Δdhgd-1* mutants compared to wild-type animals. Finally, we found that supplementation of vitamin B12 to *Δdhgd-1* mutants reduces accumulation of lysine and increases the levels of 2AA and glutarate (**S3C, S3D, S3F, and S3I Fig**) and that *hphd-1* RNAi increases abundance of glutarate and lowers lysine levels (**S3G and S3J Fig**). Taken together, our results indicate that the ketogenic amino acid lysine contributes to energy production via ketone body production and that this is impaired in *Δdhgd-1* animals, which respond by up-regulation of ketone body metabolism genes.

## A model for propionate catabolism in ketone body production and use in *C. elegans*

What is the physiological role of ketone body metabolism in *C. elegans*? In humans, ketone bodies are synthesized by the liver by the breakdown of fatty acids and the ketogenic amino acids, lysine and leucine. The ketone body precursor, acetyl-CoA is converted into ketone bodies that are transported to tissues with high energetic demands (brain and muscle) [38]. Therefore, we hypothesized that, as in humans, ketone bodies may provide an energy source in *C. elegans* to support development.

We obtained insight into the potential function of ketone bodies as an energy source in *C. elegans* by examining tissue-level RNA-seq data of larvae [39] and embryos [40]. Specifically, we discovered that the canonical propionate breakdown pathway genes are extremely highly expressed in body wall muscle, relative to other tissues (**Figs 4F and S9**). A major function of muscle metabolism is to produce energy for movement, which is critical for the developing *C.*

*elegans* embryos. The observation that the canonical propionate breakdown pathway is most highly expressed in body wall muscle therefore indicates that this SCFA may serve as an important energy source for this tissue by generating succinyl-CoA that can enter the TCA cycle to support ATP production. Together with our experimental and modeling data described above, this leads to a model where muscle tissue generates energy via the canonical propionate breakdown pathway on diets replete in vitamin B12. When dietary vitamin B12 is low, however, the expression of the propionate shunt is activated [27,28]. However, inspection of the tissue-level RNA-seq data mentioned above revealed that the first shunt gene, *acdh-1*, is not expressed in body wall muscle at all, which indicates that this pathway is not active in this tissue (**Figs 4F and S9**). Instead, this gene is most highly expressed in the animal's intestine (**Figs 4F and S9**) and, at lower levels, in the hypodermis (**Fig 4F**). In *C. elegans*, intestine and hypodermis function as both gut and liver, which are the primary sites of nutrient absorption, digestion, and metabolism [22,41]. Therefore, we hypothesized that, on low vitamin B12 diets, the propionate shunt produces acetyl-CoA in the intestine (and to a lesser extent hypodermis) and that this acetyl-CoA is converted to ketone bodies (**Fig 4G**). These ketone bodies can then diffuse to the muscle, where they are converted back to acetyl-CoA and oxidized via the TCA cycle to produce energy.

## Discussion

In this study, we identified a metabolic and physiological function of D-2HG production and recycling in *C. elegans*. This recycling facilitates flux through the propionate shunt and loss of *dhgd-1*, which encodes the enzyme that converts D-2HG back to αKG, results in mitochondrial defects and embryonic lethality. However, we discovered that these phenotypes are caused by different metabolic changes. On the one hand, mitochondrial dysfunction in *Δdhgd-1* mutants is caused by high levels of 3HP, which impairs mitochondrial health by binding IMMT-1 and inhibiting its membrane-shaping activities [30]. On the other hand, our data strongly suggest that embryonic lethality in *Δdhgd-1* mutants is caused by impaired ketone body metabolism. We propose that ketone bodies are generated from the breakdown of leucine, which produces AA and ketogenic acetyl-CoA, and lysine, which produces 3HB and, perhaps, even chain fatty acids, which generate acetyl-CoA. Our data suggest that *Δdhgd-1* animals have a reduced capacity to generate energy as determined by impaired degradation of ketogenic amino acids, reduced flux through the propionate shunt, and depletion of TCA cycle intermediates.

There are 3 different mechanisms by which embryonic lethality in *Δdhgd-1* mutant animals can be rescued, and each of these can be explained by our energy model. First, vitamin B12 supplementation fully rescues lethality without necessarily lowering 2HG levels. We propose that this is because vitamin B12 rescues embryonic energy supply by facilitating production of propionate-derived succinyl-CoA in the muscle. Succinyl-CoA can enter the TCA cycle to produce energy, thereby circumventing the need for ketone body production either by the shunt or by breakdown of lysine or leucine. Second, direct supplementation of the ketone bodies 3HB and AA partially rescues lethality, likely because it can diffuse to the muscle where it can be converted to acetyl-CoA to produce energy. Finally, RNAi of *hphd-1* also fully rescues lethality in *Δdhgd-1* mutant animals. This is a key result showing that loss of *dhgd-1* does not cause lethality solely by impairing flux through the propionate shunt, because *hphd-1* functions in the same pathway, even in the same reaction coupled with *dhgd-1*. Our metabolomic data showed that loss of *dhgd-1* may impair lysine and leucine breakdown, which may occur as a result of 2HG accumulation, which is known to inhibit a variety of metabolic enzymes, including BCAT, a key enzyme in the breakdown of all 3 BCAAs [12]. Altogether, our results indicate

that *C. elegans* may rely on amino acids and propionate as a source of energy, and that on diets low in vitamin B12, the propionate shunt is an important producer of the ketone body precursor acetyl-CoA. We have previously shown that the propionate shunt plays an important role in the detoxification of propionate, high levels of which are detrimental to *C. elegans*, as they are in humans [27,28]. Therefore, the results presented here indicate that the propionate shunt serves an important dual purpose: the detoxification of excess propionate, and the production of acetyl-CoA and energy.

## Methods

### Resource availability

**Materials availability.** *hphd-1* RNAi and vector control *E. coli* OP50 (xu363) strains are available upon request.

### Experimental model and subject details

**C. elegans and E. coli strains.** For maintenance, *C. elegans* strains were grown on nematode growth medium (NGM) seeded with *E. coli* OP50 and supplemented with 64 nM adenosylcobalamin (vitamin B12). Experimental plates contained vitamin B12 if specified. The mutant strain F54D5.12*(tm6671)* was provided by the National Bioresource Project for Nematode *C. elegans* [42] and was backcrossed 3 times to N2. We named F54D5.12 *dhgd-1* for D̲-2-hydroxyglutarate d̲ehydrogenase. Wild-type N2 *C. elegans* and bacterial strains *E. coli* OP50 and *E. coli* HT115 were obtained from the Caenorhabditis Genetics Center (CGC), which is funded by NIH Office of Research Infrastructure Programs (P40 OD010440). The RNAi-compatible *E. coli* OP50(xu363) was provided by the Xu lab [34].

### Method details

**Protein sequence similarity.** Similarity between *C. elegans* DHGD-1 and human D2HGDH was established by NCBI's BlastP reciprocal best match (46% sequence similarity). Conserved FAD-oxidase and FAD-binding domains were found in both proteins by search against Pfam database using NCBI's Protein Domain Search.

**Metabolite extractions.** *C. elegans* cultures maintained on *E. coli* OP50 diet supplemented with 64 nM vitamin B12 were synchronized using sodium-hydroxide–buffered sodium hypochlorite treatment. L1 larvae were plated onto NGM agar plates for specified treatment (e.g., vitamin B12 supplementation, RNAi) and harvested as 1-day-old gravid adults. Animals used for experiments described in Fig 1C were cultured in liquid S-medium [43] in conditions described in the isotope tracing section of the methods. First day gravid adults were washed 3 times with M9 buffer and 50 μl of the packed animal pellet was flash-frozen in a dry ice/ethanol bath and stored at −80˚C. Next, samples were mixed with 1 mL 80% methanol and 0.5 mL of 200- to 300-μm acid-washed glass beads (MilliporeSigma) and homogenized using a FastPrep-24 bead beater (MP Biomedicals), with intermittent cooling in dry ice/ethanol bath. Samples were then extracted for 15 min, then centrifuged for 10 min at 20,000 × *g*, and the supernatant was used immediately or stored at −80˚C.

**Targeted quantification of metabolites using gas chromatography–mass spectrometry (GC-MS).** Approximately 250 μl of animal extracts were dried under vacuum using a Speed-Vac concentrator SPD111V (Thermo Fisher Scientific). Derivatization of dried samples was performed by adding 20 μl pyridine and 50 μl *N*-methyl-*N*-(trimethylsilyl)trifluoroacetamide (MSTFA, MilliporeSigma) and incubating samples for 3 h at 37˚C, followed by 5 h incubation at room temperature. In the experiments that also targeted αKG, dried samples received 20 μl

**Table 1. Key resources.**

| REAGENT or RESOURCE | SOURCE | IDENTIFIER |
|---|---|---|
| Bacterial and virus strains | | |
| *Escherichia coli* OP50 | CGC | N/A |
| *Escherichia coli* OP50 (xu363) | [34] | N/A |
| Chemicals, peptides, and recombinant proteins | | |
| Coenzyme B12 (adenosyl cobalamin) | MilliporeSigma | Cat# C0884 |
| *N*-methyl-*N*-(trimethylsilyl)trifluoroacetamide (MSTFA) | MilliporeSigma | Cat#M-132 |
| Methoxyamine hydrochloride | MilliporeSigma | Cat# 226904 |
| DL-3-hydroxybutyric acid sodium salt | MilliporeSigma | Cat# H6501 |
| Propionic acid | MilliporeSigma | Cat#81910 |
| Sodium propionate (D5, 98%) | Cambridge Isotope Laboratories | Cat#DLM-1601-1 |
| L-Lysine monohydrochloride (13C6, 99%) | Cambridge Isotope Laboratories | Cat# CLM-2247-H |
| L-Lysine monohydrochloride | MilliporeSigma | Cat# L5626 |
| Ethyl acetoacetate | MilliporeSigma | Cat# 00410 |
| Ethyl 3-hydroxybutyrate | MilliporeSigma | Cat# E30603 |
| Ethyl acetate | MilliporeSigma | Cat# 270989 |
| Deposited data | | |
| Transcriptome dataset (Effect of mutated *dhgd-1(tm6671)* on *C. elegans* metabolism) | This study | https://www.ncbi.nlm.nih.gov/geo/query/acc.cgi?acc=GSE201645 |
| Experimental models: Organisms/strains | | |
| *Caenorhabditis elegans* N2 (wild type) | CGC | N/A |
| *Caenorhabditis elegans* F54D5.12(tm6671) | [42] | N/A |
| *Caenorhabditis elegans* Pmyo-3::GFPmito; Pmyo-3::lacZ::GFP(nls) | CGC | PD4251 |
| *Caenorhabditis elegans* emb-27(g48) II | Dr. J.E. Irazoqui | GG48 |
| *Caenorhabditis elegans* acdh-1(ok1489) I | CGC | VC1011 |
| *Caenorhabditis elegans* nhr-68(gk708) V | CGC | VC1527 |
| Software and algorithms | | |
| R: A Language for Data Analysis and Graphics, version 4.1 | http://www.r-project.org | N/A |
| WormFlux | http://wormflux.umassmed.edu/ | N/A |
| Other | | |
| Single quadrupole GC/MS | Agilent Technologies | 5977B GC/MSD |

CGC, Caenorhabditis Genetics Center; GC-MS, gas chromatography–mass spectrometry.

of 20 mg/mL methoxyamine hydrochloride in pyridine (MilliporeSigma) and were incubated at 37°C for 1 h before reaction with MSTFA derivatization. Measurements were performed on an Agilent 7890B single quadrupole mass spectrometry coupled to an Agilent 5977B gas chromatograph (GC-MS) (Table 1) with an HP-5MS Ultra Inert capillary column (30 m × 0.25 mm × 0.25 μm). The inlet temperature was set to 230°C, the transfer line was at 280°C, and the MS source and quadrupole were at 230°C and 150°C, respectively. The oven was programmed to start at 80°C, hold for 2 min, and ramp-up at 5°C/min until 280°C. Each metabolite was identified based on retention time, 1 quantifier, and 2 qualifier ions that were manually selected using a reference compound. Peak integration and quantification of peak areas was done using MassHunter software, blank subtraction, and normalization to total quantified metabolites were done in R software.

**Relative quantification of D- and L-2HG.** To distinguish the 2 enantiomers of 2HG, we adapted a previously published protocol [44]. First, 300 μl of *C. elegans* metabolite extract was evaporated to dryness in glass inserts. Approximately 50 μl of R-(-)-butanol and 5 μl of 12N hydrochloric acid were added to each insert and incubated at 90˚C with shaking for 3 h. Samples were cooled to room temperature and transferred into glass tubes containing 400 μl hexane. After extraction, 250 μl of organic phase were transferred into a new glass insert and evaporated to dryness. Next, 30 μl of pyridine and 30 μl of acetic anhydride were added to each sample and allowed to incubate for 1 h at 80˚C with shaking. Samples were dried, resuspended in 60 μl of hexane, and immediately analyzed by GC-MS using targeted method settings, with exception of oven ramp, which was run from 80 to 190˚C at the rate of 5˚C/min and then until 280˚C at 15˚C/min. A total of 173 *m/z* ion was used for quantification of both D- and L-2HG.

**Brood size and hatching assays.** Seven L4-stage animals per strain/condition were placed on individual 3.5 cm plates and transferred to a new plate every 24 h until animals stopped laying eggs. Plates with embryos were incubated for 24 h and then the number of hatched L1 larvae and unhatched embryos were counted and averaged for all animals. Brood counts for animals that died or crawled off the plate before egg-laying was complete were excluded. The experiment was repeated 3 times.

**Mitochondrial network imaging.** Wild-type and *Δdhgd-1 C. elegans* strains were crossed to a strain with fluorescently labeled mitochondria and nuclei *Pmyo-3*::*GFP*mito; *Pmyo-3*::*lacZ*::*GFP*(nls) in body wall muscles. Animals maintained on *E. coli* OP50 diet supplemented with 64 nM vitamin B12 were synchronized using sodium-hydroxide–buffered sodium hypochlorite treatment. Synchronized L1 animals were cultured in specified conditions until L4 stage. At least 10 L4 animals per condition were imaged using Nikon A1 point-scanning confocal microscope with 561 nm laser. Imaging was performed using an Apo TIRF, N.A. 1.49, 60× oil immersion objective in galvano imaging mode.

**Embryo collection.** *C. elegans* cultures maintained on *E. coli* OP50 diet supplemented with 64 nM vitamin B12 were synchronized using sodium-hydroxide–buffered sodium hypochlorite treatment. Approximately 25,000 L1 larvae were cultured in liquid S-medium [43] in conditions described in the isotope tracing section of the methods. First day gravid adults were treated with sodium-hydroxide–buffered sodium hypochlorite, resulting eggs were washed 5 times with M9 buffer and incubated until majority of the embryos are at comma or 1.5-fold stage. A total of 200,000 embryos per sample were flash-frozen in a dry ice/ethanol bath and stored at −80˚C until metabolite extraction.

**Gene coexpression analysis.** A ranked list of metabolic genes that are coexpressed with *dhgd-1* was extracted from a compendium of 169 expression datasets. Briefly, z-normalized expression datasets with at least 10 conditions were combined to form a global coexpression matrix. Correlation in expression between metabolic gene pairs across the compendium was calculated as Pearson correlation coefficient. Gene set enrichment analysis (GSEA) was performed on the ranked coexpression list using the PreRank module of GSEA [45] with pathway-to-gene annotations from WormPaths [35] and the significance cutoff set at a false discovery rate (FDR) of less than or equal to 0.05.

**Deuterium isotope tracing.** Gravid adults were treated with sodium-hydroxide–buffered sodium hypochlorite solution to obtain synchronous L1 populations. Approximately 50,000 L1 animals were added per 50 mL Erlenmeyer flask containing 10 mL of K-medium with modified salt concentrations (51 mM NaCl, 32 mM KCl, 3 mM CaCl2, 3 mM MgSO4) and *E. coli* OP50 pellet from a 100 mL of overnight culture. Cultures were incubated at 20˚C with shaking at 180 rpm. Sodium hydroxide-neutralized propionic acid (MilliporeSigma) or $^2$H$_5$-propionic acid sodium salt (Cambridge Isotope Laboratories) were added to the cultures when animals reached the late L4/young adult stage to a final concentration of 20 mM. *C. elegans* were

harvested at the gravid adult stage (about 12 h after supplementation), washed 3 times, flash-frozen in ethanol/dry ice bath, and stored at −80°C. Metabolite extraction, derivatization, and GC-MS methods were as described above. M+0 3HP and 2HG were quantified as *m/z* 219 and *m/z* 247, respectively. Relative isotopic enrichment was calculated as $\frac{R}{R+1} \times 100$, where R is a difference between abundances (normalized to the M+0 ion) of isotopologues in a sample with labeled propionate and a sample with unlabeled propionate supplement.

**HPLC-MS analysis.** Reversed-phase chromatography was performed using a Vanquish HPLC system controlled by Chromeleon Software (Thermo Fisher Scientific) and coupled to an Orbitrap Q-Exactive HF mass spectrometer controlled by Xcalibur software (Thermo Fisher Scientific). Extracts prepared as described above were separated on a Thermo Scientific Hypersil Gold column (150 mm × 2.1 mm, particle size 1.9 μm) maintained at 40°C with a flow rate of 0.5 mL/min. Solvent A: 0.1% formic acid in water; solvent B: 0.1% formic acid in acetonitrile. A/B gradient started at 1% B for 3 min after injection and increased linearly to 98% B at 20 min, followed by 5 min at 98% B, then back to 1% B over 0.1 min, and finally held at 1% B for an additional 2.9 min to re-equilibrate the column. Mass spectrometer parameters: spray voltage (−3.0 kV, +3.5 kV), capillary temperature 380°C, probe heater temperature 400°C; sheath, auxiliary, and sweep gas 60, 20, and 2 AU, respectively. S-Lens RF level: 50, resolution 120,000 at *m/z* 200, AGC target 3E6. Samples were injected and analyzed in negative and positive electrospray ionization modes with *m/z* range 117–1000. Analysis was performed with Xcalibur QualBrowser v4.1.31.9 (Thermo Scientific).

**Metabolite supplementation.** For experiments using vitamin B12 supplementation, NGM agar plates were supplemented with 64 nM adenosylcobalamin before pouring plates. DL-3-hydroxybutyric acid sodium salt, ethyl 3-hydroxybutyrate, ethyl acetate, and ethyl acetoacetate (MilliporeSigma) (Table 1) were supplemented in indicated concentrations to NGM agar media.

**Embryonic lethality assays.** Gravid adults were treated with sodium-hydroxide–buffered sodium hypochlorite solution, and released embryos were washed and incubated in M9 buffer for 18 h to obtain synchronized L1 animals. Approximately 30 animals were placed onto 3.5 cm NGM plates (seeded with 100 μl of overnight bacterial culture 1 day before) and allowed to lay eggs. Adult animals were washed away with M9 buffer and approximately 200 to 300 embryos were transferred onto a new plate with the same supplements. After 24 h, hatched and not hatched animals were counted.

**RNAi assays.** RNAi experiments were done using *E. coli* OP50(xu363) [34] transformed with either empty vector L4440 or RNAi plasmid. Bacterial cultures were grown 18 to 20 h and seeded onto NGM agar plates containing 2 mM isopropyl β-D-1-thiogalactopyranoside (IPTG), 50 μg/mL ampicillin, and used for metabolomics or phenotypic assays as described.

**Expression profiling by RNA-seq.** Approximately 200 to 300 synchronized gravid adults were harvested from *E. coli* OP50-seeded NGM plates with or without supplemented 64 nM adenosylcobalamin. Animals were washed 3 times with M9 buffer, and total RNA from their bodies (excluding embryos) was extracted using the RNeasy kit (Qiagen), with an additional step of on-column DNase I (NEB) treatment. RNA quality was verified by agarose gel electrophoresis.

RNA-sequencing was performed as previously described [29]. Briefly, multiplexed libraries were prepared using Cel-seq2 [46]. Two biological replicates were sequenced with a NextSeq 500/550 High Output Kit v2.5 on a Nextseq500 sequencer and 3 other replicates were sequenced on MGISEQ-2000. The libraries were first demultiplexed by a homemade python script, and adapter sequences were trimmed using trimmomatic-0.32 by recognizing polyA and barcode sequences. Then, the alignment to the reference genome was performed by

STAR. Features were counted by ESAT [47] with pseudogenes discarded. The read counts for each gene were used in differential expression analysis by DESeq2 package in R 3.6.3 [48]. Batch effects were corrected in DESeq2 statistical model by adjusting the design formula.

**Flux balance analysis.**   FBA was done using established methods [23] with the *C. elegans* metabolic network model iCEL1314 [22], which can be accessed at http://wormflux. umassmed.edu/. Before simulations, the model was modified by associating HPHD-1 solely with the oxidation of 3HP [27]. In all simulations, regular model constraints, such as allowed bacterial intake and forced growth-independent maintenance energy cost, were applied using reaction boundaries. With 2 additional constraints, glyoxylate shunt pathway was made irreversible and propionate secretion was prevented (i.e., by setting the lower boundary of reaction RM00479 and the upper boundary of reaction EX00163 as zero, respectively). A background metabolic activity was set using reaction BIO0107, which produces biomass with all possible macro-components [22], as the objective function. The flux of this reaction was maximized first without any additional constraints.

Then, to obtain maximum biomass production potential at low B12 conditions, the flux of B12-dependent methionine synthase (MS) reaction (RC00946) was constrained to half of its value and the maximization was repeated. This constraint reduced the biomass flux (i.e., the maximum flux of BIO0107) by about 9% as expected [23]. For subsequent simulations, the flux of BIO0107 was constrained to be at least at this level to represent the background metabolic activity, while the constraint on MS was removed (except for low B12 simulations, see below). FBA results were obtained using objective functions that maximized the fluxes of reactions RCC0005, EX00164, EX03197, which represent the generation of energy, and the production of acetoacetate and 3HB, respectively. For each objective, 4 FBA runs were first done using the following constraints: (i) no additional constraint was applied; (ii) DHGD-1 reaction (RM03534) was constrained to zero flux to represent the mutated *dhgd-1* and resulting removal of flux through the propionate shunt; (iii) *aass-1*, L-lysine-alpha-ketoglutarate reductase reaction (RM00716), was constrained to zero flux to represent the lack of lysine degradation; and (iv) methylcrotonyl-CoA carboxylase reaction (RM04138) was constrained to zero flux to represent the lack of leucine degradation. The constraint of each run was added in the respective order, while maintaining the constraints from previous runs. Finally, all 4 FBA runs were repeated with 2 additional constraints that represent low B12 conditions: RC00946 was constrained to half of its optimal value as described above and methylmalonyl-coA mutase reaction (RM00833) was constrained to zero value.

**[13]C Lysine isotope tracing.**   Gravid adults were treated with sodium-hydroxide–buffered sodium hypochlorite solution to obtain synchronous L1 populations. L1 *C. elegans* were added to NGM plates seeded with *E. coli* OP50. When animals reached the late L4/young adult stage, they were transferred to the K-medium liquid cultures (described above) supplemented lysine hydrochloride (MilliporeSigma) and 13C6 lysine hydrochloride (Cambridge Isotope Laboratories) to the final concentration of 6 mM. Animals were harvested at the gravid adult stage (12 h after supplementation), washed 3 times, flash-frozen in ethanol/dry ice bath, and stored at −80˚C. Metabolite extraction, derivatization, and GC-MS methods were as described above. M +0 lysine, 2-aminoadipate, and citrate/isocitrate were quantified as *m/z* 317, *m/z* 260, and *m/z* 273, respectively, at retention times determined by comparison with corresponding standards. Isotopic enrichment was calculated using R package IsoCorrectoR [49].

## Quantification and statistical analysis

All statistical details of experiments can be found in the figure legends.

## Supporting information

**S1 Fig. Structural similarities between *C. elegans* DHGD-1 and human D2HGDH.** (A) Conserved functional domains in *C. elegans* DHGD-1 and human D2HGDH according to Pfam database. (B) AlphaFold 3D models of *C. elegans* DHGD-1 (green) and human D2HGDH (blue) aligned in PyMOL. (C) Chromatograms of WT (black) and *Δdhgd-1* (red) *C. elegans* metabolite extracts compared to 20 mM γ-hydroxybutyrate standard (blue).
(PDF)

**S2 Fig. Mitochondrial defects in *Δdhgd-1* mutants.** (A, B) Representative images of mitochondria labeled with *Pmyo-3*::*GFP*mito in the wall body muscle of L4 larval stage animals. Defects in mitochondrial morphology of *Δdhgd-1* mutant *C. elegans* is rescued by supplementing vitamin B12 (A) but not by *hphd-1* RNAi (B). Scale bar 10 μm. Nuclei are marked with "n."
(PDF)

**S3 Fig. Metabolic changes caused in WT and *Δdhgd-1* animals by vitamin B12 supplementation or *hphd-1* RNAi.** (A, B) GC-MS profiling of metabolic changes in embryos of animals on *E. coli* OP50 diet: (A) *Δdhgd-1* mutants compared to WT *C. elegans*, (B) *Δdhgd-1* mutants supplemented with vitamin B12 compared to *Δdhgd-1* mutants. (C, D) GC-MS profiling of metabolic changes in L4 larvae on *E. coli* OP50 diet: (C) *Δdhgd-1* mutants compared to WT *C. elegans*, (D) *Δdhgd-1* mutants supplemented with vitamin B12 compared to *Δdhgd-1* mutants. (E–G) GC-MS profiling of metabolic changes in embryos of animals on *E. coli* OP50 (xu363) diet: (E) *Δdhgd-1* mutants compared to WT *C. elegans*, (F) *Δdhgd-1* mutants supplemented with vitamin B12 compared to *Δdhgd-1* mutants, (G) *Δdhgd-1* mutants treated with *hphd-1* RNAi compared to *Δdhgd-1* mutants on vector control. (H–J) GC-MS profiling of metabolic changes in gravid adults of animals on *E. coli* OP50 (xu363) diet: (H) *Δdhgd-1* mutants compared to WT *C. elegans*, (I) *Δdhgd-1* mutants supplemented with vitamin B12 compared to *Δdhgd-1* mutants, (J) *Δdhgd-1* mutants treated with *hphd-1* RNAi compared to *Δdhgd-1* mutants on vector control. All panels: *P*-values are Benjamini–Hochberg adjusted.
(PDF)

**S4 Fig. 3HP and 2HG measurements by GC-MS.** (A, B) GC-MS quantification of 3HP (A) and 2HG (B) in *C. elegans* and *E. coli* OP50 supplemented with propionate, $^2H_5$-propionate or untreated. Bars represent mean, each dot represents an independent biological replicate. The data underlying **S4 Fig** can be found in **S1 Data**.
(PDF)

**S5 Fig. Effect of vitamin B12 on 3HP and 2HG levels in *Δdhgd-1* mutant animals.** (A) Spearman correlation between GC-MS-quantified 3HP and 2HG levels in *Δdhgd-1* animals supplemented with vitamin B12. (B) GC-MS quantification of 2HG in WT and *Δdhgd-1* *C. elegans* fed *E. coli* OP50 (xu363) diet. The data underlying **S5 Fig** can be found in **S1 Data**.
(PDF)

**S6 Fig. GC-MS measurement of intermediates in degradation of ketogenic amino acids lysine and leucine.** (A–D) GC-MS quantification of lysine (A), 2-aminoadipate (B), HMB (C), and glutarate (D) in *Δdhgd-1* mutants and wild-type (WT) animals. Each dot represents an independent biological replicate, **$p < 0.01$, ***$p < 0.001$. The data underlying **S6A–S6D Fig** can be found in **S1 Data**.
(PDF)

**S7 Fig. Effect of 3HB on *Δdhgd-1* mutant animals.** (A) 3HB rescues lethality in *Δdhgd-1* mutants. Each dot represents an independent biological replicate and bars indicate means. (B)

Effect of 3HB and vitamin B12 on lethality of *dhgd-1*, *acdh-1*, *nhr-68*, and *emb-27* mutants. Each dot represents an independent biological replicate and bars indicate means. (C, D) GC-MS quantification of 3HB in WT and *Δdhgd-1 C. elegans* supplemented with 100 mM 3HB: (C) adults, (D) embryos. (E) Representative images of mitochondria labeled with *Pmyo-3::GFP*mito in the wall body muscle of L4 larval stage animals. Defects in mitochondrial morphology of *Δdhgd-1* mutant *C. elegans* are not rescued by supplementing 100 mM 3HB. Scale bar 10 μm. Nuclei are marked with "n." Panels A–D: the means of 3 or more groups were compared with ANOVA, followed by unpaired *t* test (*$p < 0.05$, **$p < 0.01$, ***$p < 0.001$). The data underlying **S7A–S7D Fig** can be found in **S1 Data**.
(PDF)

**S8 Fig. Isotope tracing experiment with [13]C-labeled lysine.** (A–C) Fractional enrichment of lysine (A), 2-aminoadipate (B), and citrate/isocitrate (C) isotopologues in WT and *Δdhgd-1* mutant animals fed [13]C-labeled lysine. Bars indicate mean of $n = 2$ biological replicates. The data underlying **S8A–S8C Fig** can be found in **S1 Data**.
(PDF)

**S9 Fig. Embryonic tissue expression of genes comprising the 2 *C. elegans* propionate degradation pathways.** Canonical, vitamin B12-dependent pathway (top) and propionate shunt (bottom) from a published dataset [39]. The data underlying **S9 Fig** can be found in **S1 Data**.
(PDF)

**S1 Table. Conserved domains in *C. elegans* DHGD-1 and human D2HGDH sequences.**
(XLSX)

**S2 Table. Gene sets enriched in genes coexpressed with *dhgd-1*.**
(XLSX)

**S3 Table. Genes differentially expressed in *Δdhgd-1* mutant *C. elegans* as compared to wild type.**
(XLSX)

**S4 Table. WormFlux pathway enrichment analysis of genes differentially expressed in *Δdhgd-1* mutant animals with and without vitamin B12.**
(XLSX)

**S1 Data. Underlying numerical data for Figs 1C, 1F, 2F, 2G, 3B–3E, 4B–4F, S4, S5, S6, S7A–S7D, S8A–S8C and S9.**
(XLSX)

## Acknowledgments

We thank members of the Walhout lab, especially Yong-Uk Lee, for discussion and critical feedback on the manuscript. We thank Dr. Javier E. Irazoqui for providing *C. elegans* emb-27 *(g48)* strain.

## Author Contributions

**Conceptualization:** Olga Ponomarova, Albertha J. M. Walhout.

**Formal analysis:** Alyxandra N. Starbard.

**Funding acquisition:** Albertha J. M. Walhout.

**Investigation:** Olga Ponomarova, Hefei Zhang, Xuhang Li, Shivani Nanda, Thomas B. Leland, Bennett W. Fox, Alyxandra N. Starbard, Gabrielle E. Giese, L. Safak Yilmaz.

**Methodology:** Alyxandra N. Starbard.

**Project administration:** Albertha J. M. Walhout.

**Resources:** Albertha J. M. Walhout.

**Supervision:** Frank C. Schroeder, L. Safak Yilmaz, Albertha J. M. Walhout.

**Writing – original draft:** Olga Ponomarova, Albertha J. M. Walhout.

**Writing – review & editing:** Olga Ponomarova, Alyxandra N. Starbard, Albertha J. M. Walhout.

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
