## [Editor Report · Decision Letter 0]

27 Jun 2022

Dear Dr Walhout, 

Thank you for submitting your manuscript entitled "A D-2-Hydroxyglutarate dehydrogenase mutant reveals a critical role for ketone body metabolism in Caenorhabditis elegans development" for consideration as a Research Article by PLOS Biology.

Your manuscript has now been evaluated by the PLOS Biology editorial staff as well as by an academic editor with relevant expertise and I am writing to let you know that we would like to send your submission out for external peer review.

Once your full submission is complete, your paper will undergo a series of checks in preparation for peer review. After your manuscript has passed the checks it will be sent out for review. To provide the metadata for your submission, please Login to Editorial Manager (https://www.editorialmanager.com/pbiology) within two working days, i.e. by Jun 29 2022 11:59PM.

Kind regards,

Luke

Lucas Smith, Ph.D.

Associate Editor

PLOS Biology

lsmith@plos.org

---

## [Decision Letter · Decision Letter 1]

11 Aug 2022

Dear Dr Walhout,

Thank you for your patience while your manuscript "A D-2-Hydroxyglutarate dehydrogenase mutant reveals a critical role for ketone body metabolism in Caenorhabditis elegans development" was peer-reviewed at PLOS Biology. It has now been evaluated by the PLOS Biology editors, an Academic Editor with relevant expertise, and by several independent reviewers. 

In light of the reviews, which you will find at the end of this email, we would like to invite you to revise the work to thoroughly address the reviewers' reports. As you will see, the reviewers agree that the data presented here is generally convincing, however they have also raised a number of important and overlapping concerns, including that additional experimental data is needed to strengthen the mechanistic insights provided in the study.

As a note, Reviewer 3 feels that the advance presented here is not of sufficient broad interest for the readership of PLOS Biology. After discussion with the Academic Editor and other reviewers, we do not share this concern. We think that, if you are able to thoroughly strengthen the study, with additional data, as the reviewers suggest, that the study would be within the scope of PLOS Biology.

Therefore, while we cannot accept the current version of this manuscript, we would welcome a much revised version of it. Given the extent of revision needed, we cannot make a decision about publication until we have seen the revised manuscript and your response to the reviewers' comments. Your revised manuscript is likely to be sent for further evaluation by all or a subset of the reviewers.

**IMPORTANT - SUBMITTING YOUR REVISION**

*Re-submission Checklist*

*Published Peer Review*

*PLOS Data Policy*

*Blot and Gel Data Policy*

Sincerely,

Lucas

Lucas Smith, Ph.D.

Associate Editor

PLOS Biology

lsmith@plos.org

REVIEWS:

Reviewer #1: Ponomarova et al. report that a function of F54D5.12 in C.elegans is to suppress levels of D-2-hydroxyglutarate, implying that the product of this gene is a D-2HG dehydrogenase (dhgd). They further report that activity of the gene (which they term dhgd-1) sustains the propionate shunt and enables ketone body synthesis. The paper demonstrates three distinct maneuvers to improve embryonic viability in dhgd-1 mutants: vitamin B12 supplementation (promoting the canonical pathway of propionyl-CoA metabolism), mutation of the propionate shunt enzyme hphd-1, and ketone supplementation. Overall, the paper makes a convincing case for the importance of ketone body metabolism in worm development. A relative limitation of the paper in its current form is that the authors depend heavily on transcriptomics data and FBA rather than direct evidence to infer metabolic activity, particularly with respect to ketone metabolism. Specific comments: 

1. The evidence that F54D5.12 is a 2HG dehydrogenase is strong, but with the data presented, it still seems possible that the product of this gene influences D-2HG levels indirectly. Does human D2HGDH rescue viability in dhgd-1 mutants? At a minimum, can the authors show the dhgd-1 protein sequence to demonstrate the presence of domains consistent with oxidoreductase activity? Is there homology with mammalian D2HGDH?

2. Can the authors provide more direct evidence of how ketones are produced in the worm? For example, in discussing the results in Fig. 4, the authors suggest that lysine and leucine are major sources of ketone bodies. Isotope labeling experiments using leucine or lysine, and tracking isotope enrichment in acetoacetate or 3-HB, could be used to determine the fraction of ketones produced from these amino acids. 

3. In multiple experiments with three or more groups (Figures 3 and 4), ANOVA should be used, instead of Student's t-test (Krzywinski et al. 2014 Nature Methods). In the figure legend, only t-test was described.

Minor points:

1. Can the authors comment on whether the gamma-hydroxybutyrate pathway is also active in worms, and if so whether this could be an alternative source of D-2HG?

2. In Figure 1D, why did the L-2HG levels seem to go down in dhgd-1 null animals (compare the green to the blue)?

3. In Figure 1F, the whole error bar seems to float above the grey column. Please double-check. 

4. In the main text describing Figure 2A, the authors stated "…including lower accumulation of glutamate, αKG, succinate…" This would be better stated as "abundance" rather than accumulation, as there seems to be a depletion rather than accumulation of these metabolites.

Reviewer #2: This manuscript by Ponomarova et al connects D-2-hydroxyglutarate (D-2HG) production/disposal to propionate and ketone body metabolism in C elegans. Note, this research group has done extensive prior work on the propionate shunt. The D-2HG disposal enzyme, here dubbed dhgd-1, is identified; dhgd-1 mutant worms are generated; D-2HG is elevated in these worms; there is severe embryonic lethality with failure of egg hatching. Metabolomics in dhgd-1 mutant worms demonstrates highly elevated 3-hydroxypropionate (3HP; an intermediate in the propionate shunt) and evidence of impaired leucine/lysine degradation with decrease in downstream metabolites. A model is proposed in which 3HP conversion to MSA by the enzyme hphd-1 is coupled to reduction of alpha-ketoglutarate to D-2HG (and subsequent recycling of D-2HG to aKG by dhgd-1); a very nice labeling experiment with deuterated propionate supports this model. Genetic epistasis experiments show that lethality in dhgd-1 mutant worms can be rescued by deletion of hphd-1; this results in normalization of 2HG levels but 3HP remains high. Supplementation with vitamin B12 can also rescue lethality in dhgd-1 mutant worms, but this is associated with normalization of 3HP and persistently elevated 2HG. Gene expression analyses indicate that vitamin B12 does not completely suppress propionate shunt enzymes in dhgd-1 mutant worms and that ketone body metabolism enzymes are elevated in dhgd-1 mutant worms regardless of B12 status. Metabolic simulations suggest that dhgd-1 may be important in ketone body metabolism. 

Overall, the manuscript is well written and the data are cleanly presented. The genetic experiments and phenotypes are compelling and interesting. The labeling experiment with deuterated propionate is a highlight. The use of correlation/modeling rather than experiments to connect D-2HG to ketone body metabolism is a major weakness. The following are some suggestions that could be addressed to improve the manuscript at the discretion of the editors.

Major: 

1) The manuscript would be strengthened by metabolomics assessments on dhgd/hphd double KO worms and dhgd KO worms supplemented with vitamin B12. Seeing the effects of these interventions on lysine/leucine degradation products and ketone bodies would help the reader better understand the balance between the propionate canonical/shunt and how it ties in with ketone body metabolism. At present, the connection of D-2HG/d2hgdh with ketone body metabolism is circumstantial. Presumably this would not be overly burdensome to the authors to perform similar analyses as shown in Figure 2A.

2) The discussions about the mechanism(s) of embryonic lethality are difficult to synthesize. Key results are rescue of lethality by hpdh KO with consequent reduction in 2HG levels (Fig 3); discussion here implies lethality related to elevated 2HG. Then vitamin B12 rescues lethality but 2HG remains high. Would help if the authors tried to clearly explain a hypothesis here. Is it elevated 2HG plus another factor resulting in lethality?

3) The failure of 2HG to be suppressed in the dhgd mutants supplemented with vitamin B12 is confusing. How is the 3HP level uncoupled from 2HG here? If further downstream metabolism of 3HP is somehow enhanced by B12, how does this happen without coupling to D-2HG recycling as the manuscript proposes is necessary? It would be helpful to perform labelling with deuterated propionate in this setting to see if the D-2HG is indeed coming from the propionate shunt.

4) Figure 4 is underwhelming in that mostly circumstantial evidence and modeling is provided, when relatively simple experiments mentioned above would directly confirm connection with ketone body metabolism and support the model proposed in the manuscript. 

Minor:

a) Discuss mechanism of mitochondrial dysfunction as detailed in the Zhou paper.

b) For Figure 3, should show mitochondrial morphology of dhgd/hpdh double KO worms or more clearly state whether or not Zhou paper showed this.

c) Page 9, 2nd paragraph states "This observation confirms that the propionate shunt remains active in the presence of vitamin B12…" Cannot conclude this from gene expression alone. Labelling experiment mentioned above would better support this notion.

Reviewer #3: This manuscript outlines a novel study centered on the role of the enzyme DHGD-1. The authors convincingly demonstrate the biochemical and physiological role of DHGD-1 in C. elegans metabolism and development. The authors then go on to use biochemical and genetic analysis, along with computational measures to elucidate the mechanisms by which DHGD-1 contributes to embryonic viability and mitochondrial function, arguing for a link between DHGD-1 function, the propionate shunt and ketone body metabolism.

 While there is some very nice science described in this manuscript, this reviewer does not believe the advances here constitute a broadly interesting and experimentally solid advance that warrants publication in a highly visible broadly interested journal such as PLOS Biology. This reviewer believes it would make a very nice contribution to a strong, but more specialized journal.

The main reasons for this assessment are that:

1) Even if all of the advances are accurate and well supported, the advances described here are not sufficient to comprise a significant enough and broadly interesting enough advance for publication in PLOS Biology, particularly because it is not clear how much of this study is relevant to other organisms or rather is specific to C. elegans. This reviewer does not believe it will advance a wider understanding of eukaryotic metabolism relevant to a broad audience. 

2) While the experiments showing DHGD-1 biological activity and its metabolic contribution to propionate metabolism are convincing and the rescue of dhgd-1 by three different methods is impressive, much of the mechanistic components of the argument are based on computational or database analyses and, while they make sense logically, are still rather speculative in nature. For example, much of the ketone body argument is built on prior knowledge of metabolic pathways, previously accumulated data, and speculative analyses. Solid linking of ketone bodies to worm physiology rests on the one experiment showing partial rescue of embryonic lethality by 3HB. This rescue is not particularly impressive, could mean many different things (outlined below), and really is the only experimental support for the ketone body hypothesis. Many of the other arguments made in this manuscript about the connection to ketone body metabolism are rather speculative and there are alternative possibilities to account for the experimental observations. 

Significant Experimental Issues:

1. Are the authors analyzing the right material? Mutation of dhgd-1 causes significant embryonic lethality, yet the authors isolated gravid adults for metabolic analysis. Therefore, it is not clear whether metabolite analysis primarily reflects dhgd-1 mutant adults or embryos, or both. This issue is or relevance, for example, the mitochondrial defects were shown in L4 animals, but these L4 animals are worms that have survived the embryo stage and grown up to L4, are the metabolite levels in L4 altered? Or are they only altered in embryos? These metabolite analyses should really be done with L4 larvae, as well as with gravid adults to resolve these possibilities. 

2. The rescue of embryonic lethality by 3HB is not strong enough to make the conclusions made in this paper. For example, 3HB may simply improve embryonic health via another mechanism. Because WT embryos hatch at 100%, one cannot assess whether 3HB may simply improve embryonic health. Perhaps, testing whether or not 3HB generally improves embryonic viability in a range of embryonically impaired mutants would help determine if 3HB nonspecifically or specifically improves embryonic health in dhgd-1 mutants. 

Minor Issues:

1. The authors should describe in more detail where the "co-expression" data are derived from. In other words, how did the database generate these results. 

2. Figure 3A starts with as schematic, but it is not clear where the schematic is derived from, it is a hypothesis, or an illustration? 

3. There is no explanation of where the ketone body genes shown in 4B came from, or what they are. 

4. Does 3HB rescue mitochondrial defects?

---

## [Decision Letter · Decision Letter 2]

8 Feb 2023

Dear Dr Walhout,

Thank you for your patience while we considered your revised manuscript "A D-2-Hydroxyglutarate dehydrogenase mutant reveals a critical role for ketone body metabolism in Caenorhabditis elegans development" for publication as a Research Article at PLOS Biology. This revised version of your manuscript has been evaluated by the PLOS Biology editors, the Academic Editor and the original reviewers.

As you will see below, the reviewers appreciate the work that has gone into this revision and are fully satisfied by the changes made. Therefore, we are likely to accept this manuscript for publication. However, before we can editorially accept your manuscript, we need you to address the following data and other policy-related requests in a revision that we think will not take very long.

**EDITORIAL REQUESTS: 

1) TITLE - we have been wondering if the title should be edited to be a bit more specific about the findings. If you agree (and if supported), we might suggest it be changed to something like "D-2-Hydroxyglutarate dehydrogenase regulates ketone body metabolism, mitochondrial integrity, and organismal development"

2) ARTICLE TYPE - we think that your manuscript would be most appropriate for our "Short Report" article type, and so request that you change the article type accordingly, when you resubmit your revised study. (This should not require any changes to the manuscript). For more information about PLOS Biology Short Reports, see here: https://journals.plos.org/plosbiology/s/what-we-publish#loc-short-reports

3) DATA REQUEST: You may be aware of the PLOS Data Policy, which requires that all data be made available without restriction: http://journals.plos.org/plosbiology/s/data-availability. For more information, please also see this editorial: http://dx.doi.org/10.1371/journal.pbio.1001797

a. Supplementary files (e.g., excel). Please ensure that all data files are uploaded as 'Supporting Information' and are invariably referred to (in the manuscript, figure legends, and the Description field when uploading your files) using the following format verbatim: S1 Data, S2 Data, etc. Multiple panels of a single or even several figures can be included as multiple sheets in one excel file that is saved using exactly the following convention: S1_Data.xlsx (using an underscore).

b. Deposition in a publicly available repository. Please also provide the accession code or a reviewer link so that we may view your data before publication. 

Fig 1C,F; Fig 2F-G; Fig 3B-E; Fig 4B-E,F; Fig S4; Fig S5; Fig S6; Fig S7A-D; Fig S8A-C; Fig S9;

>>Please also ensure that figure legends in your manuscript include information on where the underlying data can be found, and ensure your supplemental data file/s has a legend.

>>Please ensure that your Data Statement in the submission system accurately describes where your data can be found.

We expect to receive your revised manuscript within two weeks. 

*Published Peer Review History*

*Press*

Sincerely,

Lucas

Lucas Smith, Ph.D.

Associate Editor,

lsmith@plos.org,

PLOS Biology

Reviewer remarks:

Reviewer #1: The authors have addressed my comments and I believe the paper should be published.

Reviewer #2: The authors have addressed this reviewer's suggestions and critiques. The manuscript is substantially improved and ready for publication.

Reviewer #3: The reviewers have done a nice job addressing all of the most important concerns. I would recommend publication as submitted.

---

## [Editor Report · Decision Letter 3]

28 Feb 2023

Dear Dr Walhout,

Thank you for the submission of your revised Short Report "A D-2-Hydroxyglutarate dehydrogenase mutant reveals a critical role for ketone body metabolism in Caenorhabditis elegans development" for publication in PLOS Biology, and thanks also for addressing our last editorial requests in this revision. On behalf of my colleagues and the Academic Editor, Heather Christofk, I am pleased to say that we can in principle accept your manuscript for publication, provided you address any remaining formatting and reporting issues. These will be detailed in an email you should receive within 2-3 business days from our colleagues in the journal operations team; no action is required from you until then. Please note that we will not be able to formally accept your manuscript and schedule it for publication until you have completed any requested changes.

PRESS

Sincerely, 

Lucas Smith, Ph.D., Ph.D.

Associate Editor

PLOS Biology

lsmith@plos.org